# Feasibility of two screen media reduction interventions: Results from the SCREENS pilot trial

**Martin G. B. Rasmussen****\*, Jesper Pedersen**, **Line Grønholt Olesen, Peter Lund Kristensen, Jan Christian Brønd**, **Anders Grøntved**

Research Unit for Exercise Epidemiology, Department of Sports Science and Clinical Biomechanics, Centre of Research in Childhood Health, University of Southern Denmark, Odense, Denmark

\* mgrasmussen@health.sdu.dk

## Abstract

### Background

Advancements in screen media devices has transformed the way families engage with screen media. Although these modern devices offer many opportunities, e.g. communication and research online, an in-depth understanding of how these devices affect our health, is lacking. Before a definite randomized controlled trial, the SCREENS pilot study was conducted to assess compliance to and feasibility of two interventions, a measurement protocol, and a survey-based recruitment strategy. Also, the potential of the interventions to impact leisure time spent non-sedentary in children six-to-ten years of age was explored.

### Methods

Families (N = 12) were recruited through a population-based survey sent out in October of 2018 to adults (N = 1,675) in the Municipality of Middelfart, Denmark. Families were randomized to one of two two-week interventions; an Evening Restriction intervention (no screen media use after six pm) and a General Restrict intervention (limit entertainment-based screen media to three hours/week/person). Intervention compliance was assessed objectively by measuring household TV usage, smartphone and tablet activity via an application, and via screen media diaries. During baseline and follow-up, as part of larger protocol, family members wore two triaxial accelerometers for seven consecutive days. The potential of the interventions to impact non-sedentary time was explored based on means and standard errors (SEs).

### Results

Despite almost 85% and 75% reductions in leisure screen media use 0% and 50% of families were compliant in the Evening Restrict group and General Restrict group, respectively, based on strict a priori criteria. Participant feedback indicated that the General Restrict intervention generally was feasibly. Compliance to the accelerometry wear protocol was high (median non-wear was <1 hour/week). Moreover, the recruitment strategy was implemented

**Data Availability Statement:** Relevant dataset publicly available on the data repository FigShare.

DOI: https://doi.org/10.6084/m9.figshare.16917691.v1.

**Funding:** MGBR, JP, AG and LGO were funded by a European Research Council Starting Grant (no. 716657). Website: https://erc.europa.eu/. The funders had no role in the design of the study design, nor did they play a role in the collection, management, analysis, and interpretation of the data from the study. Nor did they have a role in the writing of the study protocol as well as in the decision to submit the report.

**Competing interests:** The authors declare that they have no competing interests.

and was feasible. The General restrict intervention might increase children's non-sedentary time (mean (SE): 36.6 (23) min/day, N = 6).

## Conclusions

The General Restriction intervention, the accelerometer wear protocol and recruitment strategy, appeared feasible.

## Trial registration

NCT03788525 at https://clinicaltrials.gov [Retrospectively registered; 27th of December, 2018].

## Introduction

In modern society, screen-based media devices are an ever-increasing part of everyday living. Recent advances in technology allow almost anyone to carry, in their pockets, unlimited access to practical information, communication platforms, games, and news outlets. Also, population-based survey data suggests that children [1, 2] and adults [3] of the 21st century spend much of their pastime using some form of screen-based media. Although on its face the availability of these modern devices appears to introduce improvements to everyday life, e.g. via enhanced knowledge acquisition and possibilities for immediate communication, the health implications of these behavioral shifts are not clearly understood. A concern could be that increased screen media use decreases habitual physical activity, which, in turn, might increase lifelong risk of poor well-being, morbidity and mortality.

To our knowledge nine randomized controlled studies [4–12] have investigated the impact of decreasing screen media use on physical activity habits in children. Of these, only one study found a significant increase in physical activity as a result of a screen-time reduction intervention [8]. In this study, however, no non-treatment control group was included and physical activity was measured using a self-report instrument [8], which may introduce outcome measurement bias. Of the above-mentioned trials several included small samples [4, 5, 7, 8] and, importantly, only one of these trials measured screen media use via objective means [9] putting into question the extent of adherence. Also, prior trials appear to have been designed to study the effectiveness of a screen use reduction program rather than its efficacy, and an effective intervention can appear to be ineffective if the adherence is poor (i.e. bias due to deviations from intended interventions). Lastly, these studies have traditionally assessed the effect of intervention on children's physical activity levels, where only one study also investigated the impact on caregiver physical activity levels [6]. Moreover, while findings from screen media use reductions interventions suggest an improvement in total sleep time, sleep time has almost exclusively been assessed via self-report, and because assigned screen reduction intervention is unblinded this influence participant-reported sleep time and cause outcome measurement bias [13, 14]. Also, although a meta-analysis of cross-sectional studies suggest a positive association between screen media use and stress, experimentally designed studies are needed to establish causal links [15]. Furthermore, a limitation of the existing research is the assessment of stress via self-report, and research is needed in which stress is assessed via objective means. Measurement of heart rate variability [16] and cortisol awakening response- and diurnal slope based on saliva sampling [17] are viable candidates for this purpose. Overall, there are still

clear gaps in our understanding of how this modern-day behavior affects the habitual physical activity and sleep of children and adults, and physiological stress in adolescents and adults.

To address these shortcomings, it is paramount that methodologically sound interventions are developed, whose findings may improve our understanding of the effect of screen media use on health and behavior. By conducting studies under real-life circumstances, the ecological validity of the research findings could be high. Experimental studies which reach a healthy compromise between free-living and controlled conditions, to assure compliance, would be instrumental in terms of advancing this research field. Also, as the technological advancement of screen media devices continues, it is necessary to conduct up-to-date research, which takes place in the current screen media use environment. The SCREENS pilot study was set out to address the shortcomings included in published research acknowledged above. The purpose of this paper is to describe the feasibility of the SCREENS pilot trial.

### Specific objectives

The primary objective of the current paper is to describe the degree of objectively and subjectively assessed compliance to two trial interventions. Secondary objectives were to i) describe compliance to physical activity measurements (accelerometry) in the context of an entire measurement protocol, ii) examine the feasibility of a strategy to recruit participants, and iii) to explore the potential of the interventions to impact six-to-ten-year-old children's leisure non-sedentary time (the planned primary outcome in the definitive full-scale SCREENS randomized controlled trial).

## Materials and methods

### Trial design

The SCREENS (not an abbreviation) pilot study was a two-arm parallel-group randomized feasibility trial with no control group. The reporting of the current paper was done according to the CONSORT extension for randomized pilot and feasibility trials (S1 Checklist) [18]. The pilot trial was registered at clinicaltrials.gov (Identifier: NCT03788525) in late December of 2018, three months before the completion of the trial (due to a busy work schedule). At the time of the registration only three out of 12 families had participated in the study.

### Participants, eligibility criteria, and settings

The study was conducted in free-living among participants residing in the Municipality of Middelfart in the Region of Southern Denmark. It was initiated in October of 2018 and finalized in March of 2019. The strategy of sampling of participants into the study was designed to optimize the possibility to generalize (or better judge generalizability of) the study findings in a future definitive trial to a well-defined source population (Danish families with children aged 6-10-years of age from the general population). A list of one randomly selected adult and one randomly selected child (between six and ten years of age) from all households in the Municipality of Middelfart was gathered. The list was obtained from the Danish National Civil Registry via the National Health Data Authority with the only restriction being sharing an address and child age. A survey was sent out on the 26th of October in 2018 to all adults, whose households met the above criteria (N = 1,686). At the time, the total population of the Municipality of Middelfart was N = 38,363, including N = 30,465 adults [19]. The adults received the survey in their personal e-boks, which is an electronic mailbox system available to Danish citizens from 15 years of age. The survey included an inquiry of screen media use habits of children and adults and the screen media home environment. On the final page of the survey there was

an invitation to participate in the SCREENS pilot study. Fig 1 below is an illustration of the flow of participants from sending out the survey to being included in the statistical investigations.

Six-to-10-year-old children and their parents were included in the trial. We limited our sample to children of this age, as we reasoned that a large screen media use reduction would most likely be feasible among children of this age. The aim of a future definite trial study was to examine the efficacy of limiting screen use, rather than the pragmatic effectiveness of a screen reduction intervention program. The intervention therefore had to be designed to maximize adherence to limiting recreational screen media use. One would expect that type and amount of screen media usage would be more under parental control or supervision among 6-10-year compared with older children, which we expected would influence the ability to succeed with adherence to screen use reduction. Also, we hypothesized that a large amounts of screen media use at this age displaces a significant amount of unstructured physical activity, such as free active play. Siblings four and five years, and siblings between 11–17 years, were deemed 'passive participants' and were not requested to conform to the screen media reduction intervention and measurement protocols but were requested not to use screen media devices in areas in the home shared with the rest of the family.

Families were initially eligible to participate in the trial, if they met the following inclusion criteria, assessed via questions included in the survey:

- The adult respondent had to be above the $50^{th}$ percentile for weekly screen media use amount. In absolute terms, this amounted to consuming above 2 hours and 43 minutes of leisure screen media use on a typical day of the week (based on all survey respondents). For practical reasons pertaining to recruitment, we chose to limit this criterion only to adults, whose screen media use might be a decent proxy for the household screen media use consumption.

- Households should not include children less than four years of age. This was to avoid disturbance of data collection on sleep quality and duration (an included outcome measurement) because of infants and toddlers with irregular sleep habits.

Furthermore, families had to meet the following secondary inclusion criteria, which was assessed during telephone screening:

- At least one adult and one child between six and ten years of age had to participate in the study

- All actively participating family members had to be able to remove both leisure- and work-based screen media use during evenings and during weekends

- Families had to consider the extent of their screen media usage an issue and report to be motivated to decrease leisure screen media use for a short time-period

- Lastly, the families had to declare that passive participants would respect the intervention conditions that those family members, who were active participants, had to follow

We excluded participants during telephone screening based on the following criteria:

- If children only resided part-time in the household

- If any participants had been diagnosed with stress or a sleep disorder by their general practitioner within the last 12 months

- If adults regularly worked night shifts

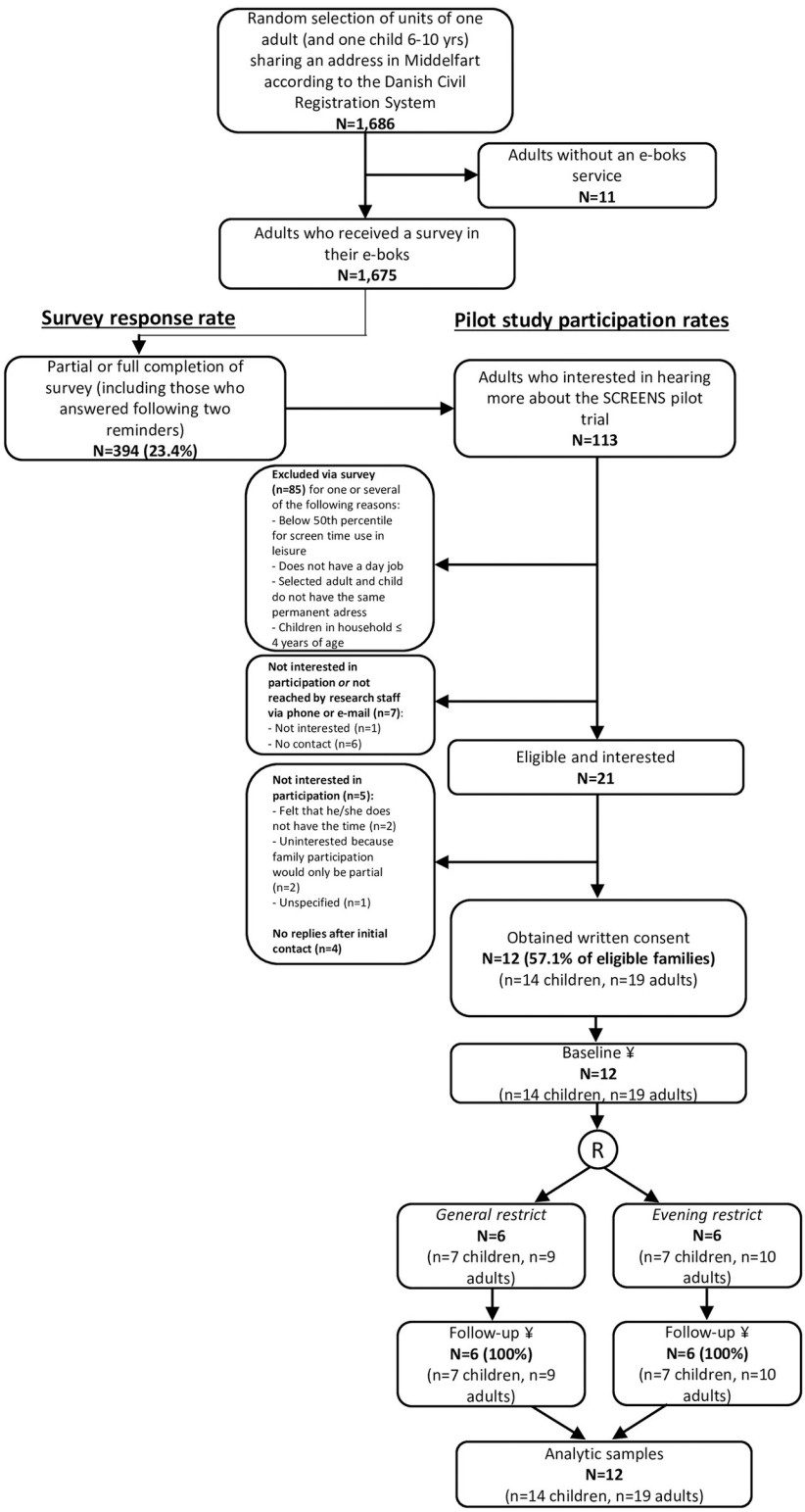

**Fig 1. Flow chart of participants from sending out a survey to being included in statistical analyses.** The flow chart above gives an overview of the steps of recruitment (electronic survey, phone contact, and meeting in households), SCREENS pilot trial participation and, finally, statistical analyses. Because the main goal of the study was to assess degree of compliance, families where included in the statistical analyses simply by not dropping out. Baseline and follow-up completion refer to objective measurement of screen media use and accelerometry. R; Randomization, ¥; Possible source of missing data.

- If family members were unable to do basic physical activity during everyday life

- If family members were diagnosed with or in the process of getting cleared from developmental disorders, such as autism spectrum disorders, or neuropsychiatric disorders, such as attention deficit hyperactivity disorder

- Lastly, if family members were already actively participating in a research study, e.g. the Odense Child Cohort study (an ongoing population-based birth study in Odense, Denmark)

Following the screening process, a mandatory meeting was held at the families' household with the member of the research team, who later would manage prescription of intervention and supervision of data collection.

The study was approved by the Regional Scientific Committee of Southern Denmark (Project-ID: S-20170213 CSF). Before participation in the study, a mandatory meeting was held in the families' household, where both verbal and written information about the study was given. Signed written consent forms had to be filled out before the study could be started. If children showed any signs of dissent, we would not proceed. However, we did not observe signs of child dissent. All data collected was stored under conditions compliant with the General Data Protection Regulation (GDPR). A more elaborate explanation of data management can be found under "Study documents" at https://clinicaltrials.gov (NCT03788525).

## Scheduled meetings during the trial

The initial trial meeting structure consisted of four meetings spanning approximately three weeks. First, a baseline start meeting, at which instructions were delivered regarding the measurement protocol at baseline and follow-up. Also, at this meeting the set-up necessary for objective measurement of screen media use in the household was established. A second meeting was held immediately following baseline assessment (a week later). At this meeting we randomized families to one of two interventions (both two weeks in duration), using alternating blocks of two or four families, stratifying according whether the child was an only child. The randomization was performed using a researcher service available in the Region of Southern Denmark; Odense Patient Network Randomize. The random allocation sequence was created by staff at Odense Patient Network Randomize and was concealed to the researchers. At a third meeting, which took place midway through the intervention, the researchers handed over equipment for follow-up assessment, taking place during the final week of the intervention. Lastly, at a final meeting, immediately following the intervention and the follow-up assessment, the researchers gathered all equipment, and handed out diplomas to the children and gave a 500 DKK reimbursement as a thank you for the families' participation. All meetings were carried out with the same researcher and took place in the families' household.

The researchers realized that the number of meetings could be decreased from four to three. The third meeting only consisting of handing out equipment for follow-up measurements, which could be done at an earlier meeting. The third meeting was substituted for a phone call, wherein the researchers had a short conversation with an adult family member, with the aim of motivating the families to be compliant to the intervention. This change in structure was implemented approximately midway through the study in a total of seven out the twelve families. The final meeting structure is shown below (Fig 2).

## Interventions

The trial included two intervention arms, whose common goal was to decrease leisure screen media use for recreational-based purposes during a two-week period. The interventions did not target work-related screen media use. In the one intervention, families had to remove all

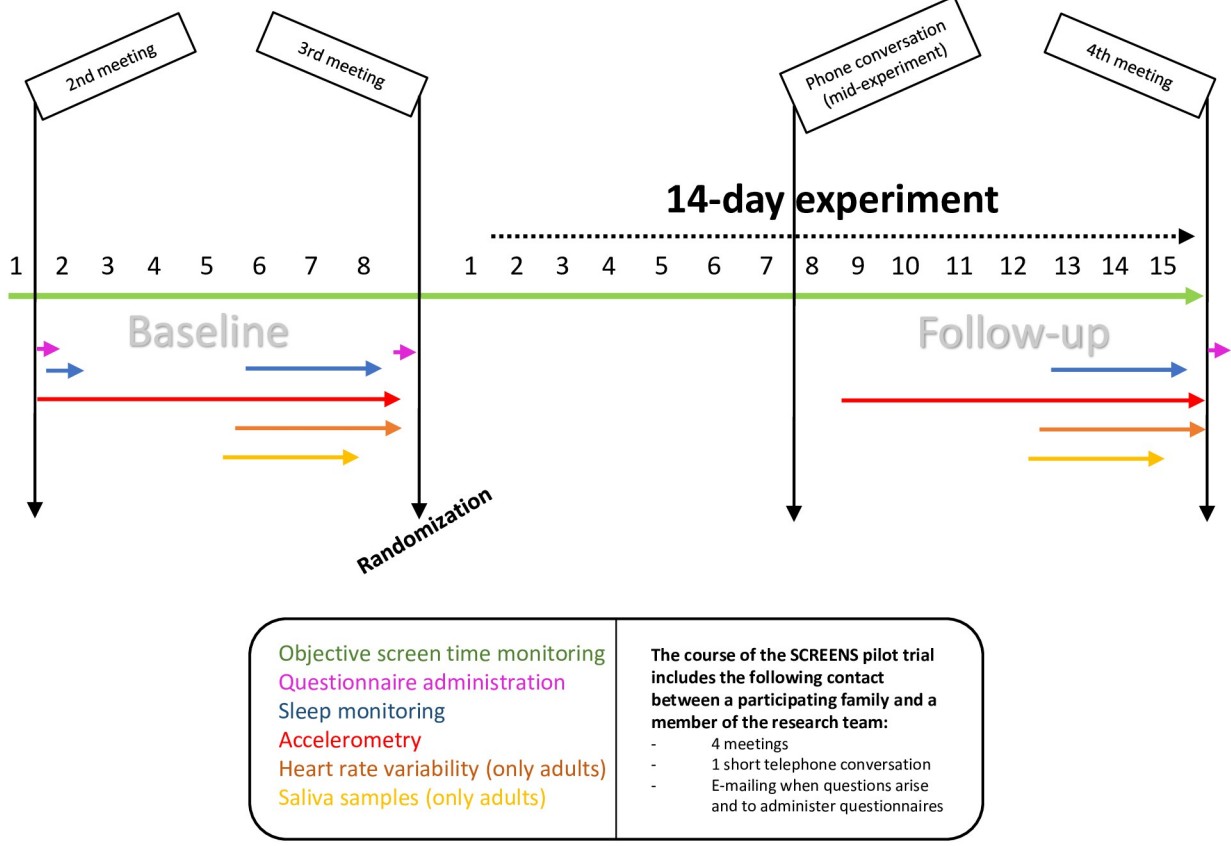

**Fig 2. Meeting schedule and measurement protocol.** The figure above illustrates the final structure of the trial in days regarding scheduled meetings, as well as the timing of exposure and outcome measurements. The program is structured such that baseline and follow-up commence on the same day of the week, so the data will be collected on the same days. Note that the only difference between baseline and follow-up is that at baseline a test sleep measurement is included during the first night. The 1st meeting (not shown above) is a mandatory information meeting, prior to the trial. Also, adults completed questionnaires at baseline addressing; mental well-being, mood state, and bodily pain and discomfort. Following baseline, a questionnaire was administrated regarding the families' experience with the baseline measurement protocol. The figure above is re-used from another publication (see Acknowledgments).

leisure screen media use, for any purpose, after six pm, on all days of the intervention. There were no restrictions on the amount of screen media use permitted before this time. In the second intervention a general screen media restriction was tested. A major component of this arm was participants had to hand over all household smartphones and tablets for the two-week period. In exchange we handed out Nokia 130 phones (not smartphones), which can only perform basic tasks such as texting, calling, alarm-setting etc. We decided to take such measures as we believed that, if the families were in possession of their screen media units, it might be a greater challenge to adhere to the intervention. However, adults in some families argued that they for practical reasons could not handover their personal phones and had to keep these during the study. A total of four adults in three families kept their smartphone. Beyond handing over the units, each participating family member had to limit their leisure screen media use on remaining devices to no more than three hours/week. Because some screen media use is necessary for parents during everyday life, e.g. to check messages from their children's teachers at online school fora and to do online financing, 30 minutes per day of such screen media use was permitted. In both interventions three 'intervention reminders'

(A5 cardboard sheets) were placed in the household next to common screen media devices, and in a common room, such as the kitchen. The sheets specified the rules of the intervention and included a picture of joyful children. These served as a motivational factor and as an environmental cue to modify screen media use behavior.

The families were instructed about the rules of the interventions both in writing and verbally. The families did not receive any written or verbal instructions or suggestions to alternative activities that could replace the families use of recreational screen media.

The theoretical basis of the interventions is described in detail in the study protocol for the randomized controlled trial [20]. Briefly, the behavioral model Social Cognitive Theory [21] was used as a theoretical foundation in the development of the interventions. The connected and reciprocal links between a person's environment, personal factors and his or her behavior [21], was used as a framework while developing the intervention components. This included objectively monitoring household screen media use and placing intervention reminders in the household (changes to the environment), in attempt to influence the person (personal factor) and assessing whether this influenced health behavior (recreational screen use).

## Outcomes (primary outcome and secondary outcome measures)

**Measurements of compliance to recreational screen media use reduction (primary outcome).** The primary outcome of the feasibility trial was compliance to the prescribed recreational screen use reduction intervention. We installed a smartphone and tablet application (SDU Device Tracker (SDU DT)) on all devices that 'active participants' were users of. We designated users to each device during installation. The application monitored whether a unit's screen was lit (timing and duration) on a second-to-second basis. No information regarding the content of usage was recorded. A detailed description of SDU DT has been described elsewhere [20]. The application is currently undergoing validation and preliminary results indicate that the application in most instances is a valid measure of total daily time of smartphone and tablets usage. However, the researchers noted three instances where the application was installed, where no or very little data was received on the university server. We also installed TV monitoring devices on all TVs in the household that 'active participants' were users of. Because this device simply measures usage of a TV, it is not user specific. The monitor uses a hall sensor to gauge the electrical current to a TV, assessing whether the TV is turned on. The voltage measured is then converted using analog to digital conversion via a micro-controller installed, using one-minute epochs. Based on the electrical signal we were able to separate the TV signal into; usage, stand-by signal and if the monitor had been shut off. We were also able to detect if the monitoring device had been unplugged from the socket (absence of a signal). A more detailed description of the TV-measurement units is described elsewhere [20]. We included active monitoring of screen media use using SDU DT and the TV monitoring device throughout the entire trial (Fig 2). We included no objective monitoring of stationary and portable pc, i.e. laptops.

Both intervention groups were handed a single sheet to report if they did not comply with the intervention. This included if they were non-compliant while using screen media devices that we did not monitor, e.g. at friends or families' house. In the General Restrict group, there was a need for an additional sheet, in which each active participant could report the amount and timing of usage, of the three hours/week of entertainment-based screen media, which was permitted. No sheet was included in which necessary screen media use could be reported.

We computed degree of compliance to the interventions based on the objective and subjective screen media use data. For the Evening Restrict group any screen media use after six pm was classified as non-compliance. We cross-referenced TV-usage reported after six pm with

TV-monitor data after six pm, to personalize non-compliance. Of the nine TVs we monitored in the six families randomized to this group, three were placed in children's room and were therefore, by default, personal. We only registered data on one shared tablet, in one family, where less than a second's activity was registered and only at baseline. Therefore, personalization of screen media usage on shared tablet devices was not possible given the sparse data.

For the General Restrict group more detailed subjective data was available. In this group, only one shared tablet was registered, which was handed over, as part of the intervention. Therefore, personalization of screen media usage on shared devices during the intervention was only possible for TVs. Each subjective reporting of TV usage was investigated and compared to registered household TV-activity. TV-activity registered within up to five hours relative to what was reported in individual sheets, which approximated the amount of TV time reported, was allocated to said user. Five hours was chosen arbitrarily, as we expected most reporting in sheets would be done retrospectively, sometimes by children, and therefore some flexibility, in terms of timing of reporting, was needed. All TV usage, which could not be personalized, was classified as residual TV-time. In the General Restrict arm, there were no personal TVs in any household. Also, in the sheets, whenever reported, we were able to detect any screen media usage outside the household (i.e. not reflected in our objective data), as well as any necessary screen media use. However, only two instances of necessary screen media use (on smartphones) were reported.

Any TV-activity, which was reported as usage solely by 'passive participants' was deleted from the TV data (set to TV-standby signal). This was done in three cases. There was also one instance, where a participant accidently unplugged the TV monitoring device and plugged in their work laptop, instead. This too was set to stand-by TV signal.

We computed the total amount of entertainment-based screen media usage during the intervention for everyone in the General Restrict group, based on the following formula:

**Amount of entertainment-based screen media use** $=$ total objectively measured screen media use (hrs/2 weeks) $+$ self-reported entertainment-based screen media use beyond objective measures (hrs/2 weeks) $-$ self-reported necessary screen media use(hrs/2 weeks) in objective measures (if $\leq 30$ minutes/day)

We then computed the proportion of this amount relative to the total amount permitted (three hours/week * two weeks), i.e. the proportion relative to the compliance threshold, as the degree of compliance.

We were not able to evaluate individual level compliance in four individuals (three from the timed group and one from the General restrict group), all of whom were children, as there for these persons were no screen media use recorded, either objectively or subjectively.

To consider that there also was TV-time in each household, which could not be personalized (residual TV-time), we also summarized compliance on a family level. We computed the total household entertainment-based screen media use output and computed the proportion relative to the permitted entertainment-based screen media use, for the whole family (n participants x three hours/week x two weeks).

We were not able to personalize baseline TV-usage as no subjective reporting was done at this time.

**Secondary outcomes (tentative outcomes in the definitive trial).** Fig 2 gives an overview of the outcome measurement protocol that we planned to include in the definitive trial. We included multiple outcome measurements in the protocol at both baseline and follow-up. We included accelerometry for a one-week period (seven x 24 hours). The baseline and follow-up

measurement of accelerometry period spanned eight days (the same days of the week twice during two separate weeks), including two weekend days, e.g. from Tuesday five pm to the following Tuesday five pm. Details regarding equipment and protocols for measurements other than accelerometry can be found elsewhere [20]. Briefly, participants collected saliva samples (three times in the morning and one time immediately before bedtime) on three consecutive days for later cortisol and cortisone assessment. The participants' heart rate variability was measured for three consecutive days using the Firstbeat Bodyguard 2.0 device. Data from saliva sampling and heart rate variability measurements were included as indicators of physiological stress. Finally, we measured sleep using the Zmachine electroencephalography-based sleep monitoring system, for three consecutive nights. At baseline, we included an extra night of sleep measurement to get acquainted with the sleep protocol. Baseline and follow-up measurements were initiated on the same day of the week, whenever possible, to ensure comparability.

The planned primary outcome in the definitive full-scale SCREENS randomized controlled trial was children's leisure non-sedentary time. A secondary outcome of the present study was therefore to examine compliance to physical activity measurements (accelerometry) in the context of the entire measurement protocol. Compliance to sleep assessment and physiological stress are reported elsewhere [22]. We employed accelerometry to objectively assess non-sedentary time and other physical activity measures using two Axivity AX3 (Axivity Ltd., Newcastle upon Tyne, United Kingdom) accelerometers, which measure acceleration in three planes. The accelerometers were worn in elastic belts around the hip and thigh for seven consecutive days at baseline and follow-up. The devices sampled at 50 Hz and the sensitivity of measurement was set to +/- eight g. The subjects were instructed only to remove the devices during water activities.

Based on algorithms developed by Skotte et. al. 2014 using a thigh-worn accelerometer in one-second epochs [23], we developed child and adolescent specific thresholds for the categorization of accelerometry into distinct daily body positions and physical activities (lying down, sitting, moving, standing, biking, running, and walking). The categorization via the algorithm into these activities based on these algorithms indicate high sensitivity and specificity ($\geq$85.8% in all cases) in children similar in age to our study population [24]. We defined non-sedentary time as any activity not in a lying or sitting position (also including standing). We decided post hoc (a priori of knowledge of data in the ongoing full-scale SCREENS randomized controlled trial) that a valid day of measurement should not include >10 percent non-wear. Furthermore, we decided that a complete measurement period at baseline and follow-up should include at least four weekdays and at least one weekend day. To be compliant the data had to be complete at both baseline and follow-up. Non-sedentary time was summarized for all valid days then divided by number of valid days in those who met our compliance criteria, to get daily amounts, at both time points.

The identification of non-wear has been described elsewhere [20]. Briefly, using data on acceleration, temperature (individually estimated non-moving temperature) and predefined child awake time (06:00 AM to 10:00 PM), periods where the belts were not worn, were identified. During baseline and follow-up measurement, participants filled out a daily checklist, where schedule information (time of awakening, when arriving at and leaving work or school, as well as bedtime) was reported. Based on checklist data we were able to time annotate the accelerometry data (as well as SDU DT and TV data) and restrict to only leisure and awake time. A description of the handling of missing schedule data can be found elsewhere (under "Study documents" at registration NCT04098913 at clinicaltrial.gov). We then computed hours of non-wear at baseline and at follow-up, as well as the number of non-wear bouts (number of sessions where non-wear time was accumulated). Then the proportion of non-wear during each day was computed. The software OmGUI version 1.0.0.37 was used to set-

up, extract, re-sample and convert the data. Raw accelerometry was processed using Matlab (Mathworks Inc., Natick, Massachusetts, US) release R2019a version 9.6.0.1099231.

In the survey used to recruit participants into the trial, highest educational attainment was obtained. Based on this information we categorized the individuals according to the International Standard Classification of Education.

**Justification of sample size.** The sample size of 12 families with an expected 1–2 children and 1–2 adults from each family participating was selected based a professional judgment by the researchers on how many families from our target population were needed to evaluate the primary and secondary outcomes of the feasibility study. Because we had no aim to incorporate hypothesis testing in the feasibility trial, we did not sample participants based on an a priori power calculation [18].

**Randomization (random number generation, allocation concealment, implementation).** Families were randomized in 1:1 allocation ratio to one of two interventions using random permuted blocks of two or four families, stratifying according whether the child was an only child. The randomization was performed using a researcher service available in the Region of Southern Denmark; Odense Patient Network Randomize. The random allocation sequence was created by staff at Odense Patient Network Randomize and they were not involved with recruitment of participants. The sequence was concealed to the researchers, and when the individual family was randomized, it was done by a researcher in the home of the participants by logging on to the Odense Patient Network Randomize web application.

**Blinding.** The study was open label due to the nature of the intervention. Thus, neither the participants nor the researchers were blinded to intervention groups. Accelerometry (the methodology to assess the tentative primary outcome in the definitive trial) data reduction was handled by a researcher that was blinded to group allocation. Otherwise, the researchers who assisted the participants in collecting the data were not blinded.

## Statistical methods

The analysis of data for the feasibility study was more descriptive in nature and no analysis of efficacy was carried out in accordance with CONSORT extension to randomized pilot and feasibility trials [18]. Baseline characteristics of the sample was computed using means and standard deviations when the data was normally distributed, medians and interquartile ranges when skewed, and proportions when the data was categorically scaled. Because some data at baseline could only be presented at a household level, baseline characteristics is presented on an individual level, and on an aggregated family level.

Box plots was created to illustrate degree of compliance to accelerometry, for the whole group and within each intervention group, separately for baseline and follow-up. We computed mean changes with standard errors (SEs) for non-sedentary time (the primary outcome of the future definitive trial) between baseline and follow-up in children and adults within each intervention group.

All data handling and computations were conducted using Stata IC 16 (Statacorp).

## Results

### Flow of participants

Fig 1 illustrates the flow of participants into the study. Of 21 eligible families, who we were able to reach either via phone only or via both phone and at a personal meeting, more than half agreed to participate. Via the proposed recruitment strategy, the goal of including 12 families in the trial was reached. As illustrated, no families, or adults or children within families, dropped out of the study. All 33 adult and child participants took part in the allocated

intervention, and in accelerometry assessment. Therefore, none were excluded from the analyses of compliance to the interventions and compliance to accelerometry. Families who completed the feasibility trial were similar in terms of sociodemographic characteristics and the amount of screen media use the children consumed compared to those who were eligible for the trial and who completed the survey (S1 Table). The screen media use amount among the children who participated in the trial appeared higher compared with the survey respondents who were ineligible for the trial and completed the survey.

## Baseline characteristics

The baseline characteristics of the twelve participating families can be found in Table 1.

Based on this very small sample, the groups appeared to be relatively well-balanced according to age, gender distribution, and highest educational attainment. As expected, the sample was homogenous in terms of age and family size. Smartphone and tablet usage, as well as TV consumption, was higher in the Evening Restrict group in both children and adults, relative to the General Restrict group. These differences were, however, less pronounced or removed when presented at an aggregate, family level. Individuals in the General Restrict group appeared to accumulate slightly more non-sedentary time.

## Intervention compliance

Based on the individual and family/household assessed compliance rates, no participants allocated to the Evening Restrict group were compliant to this protocol. Contrarily, in the General Restrict group, on the individual level the majority were compliant, whereas on a family/household level, half of the families were compliant. However, when comparing total leisure screen media use output at baseline (Table 1) and total leisure entertainment-based screen media use during the intervention (Table 2), the families in both groups markedly reduced their intervention-targeted screen media use; on a weekly basis, families in the Evening Restrict group decreased their median screen media use output after six pm by almost 85% (from median (IQR) hours/week at baseline: 26.9 (16.2, 46.8) to median (IQR) hours/week during the intervention: 4.3 (2.9, 5.8)), and families in the General Restrict group decreased their median leisure screen media use output by almost 75% (from median (IQR) hours/week at baseline: 34.4 (20, 45.9) to median (IQR) hours/week during the experiment: 9.1 (5.3, 11.6)).

## Compliance to accelerometry

Fig 3 displays the number of hours of accelerometer non-wear at baseline and at follow-up, for each intervention group.

The number of hours of non-wear according to Fig 3 suggest very high degree of compliance to accelerometry. At either measurement period, in either intervention arm, median hours of non-wear per week was less than 0.91 hours or 55 minutes. Extreme values for hours of non-wear for the Evening Restrict group at follow-up is explained by participants in one family putting on belts a day too late. Table 3 summarizes the numerical values illustrated in Fig 3, as well as provide additional descriptive information on non-wear.

Both the number of hours and number of bouts of non-wear appeared to increase slightly from baseline to follow-up, in both groups. At baseline, only six individuals had five valid weekdays and the rest (N = 28) had six (the maximum). At follow-up, one person had zero valid weekend days and one person had one valid weekend day. The rest (N = 30) had two valid weekend days (the maximum). In summary, only at follow-up and only in the General Restrict group was compliance not 100%. This was due to one child not meeting the criteria for a valid measurement period.

**Table 1. Baseline characteristics of sample and screen media use during baseline (one week).**

| | Individual level (adult or child) | | |
| --- | --- | --- | --- |
| | Evening Restrict (N = 17) | General Restrict (N = 16) | Total (N = 33) |
| *Adults* | | | |
| N | 10* | 9* | 19* |
| Age, yrs | 42 (38–45) | 45 (41–46) | 42 (39–46) |
| Gender, % female | 60 | 55.6 | 57.9 |
| Educational attainment—ISCED, % ($\leq$2, 3–5, $\geq$6) | 10/30/60 | 0/55.6/44.4 | 5.3/42.1/52.6 |
| Smartphone and tablet usage during leisure, hrs/week | 13.6 (8.1–15.4) (N = 9) | 7.5 (6–14) (N = 8) | 9.9 (7.2–15.4) (N = 17) |
| Smartphone and tablet usage after six pm, hrs/week | 6.2 (2.7–6.6) (N = 9) | 4.2 (2.8–5.9) (N = 8) | 4.6 (2.7–6.4) (N = 17) |
| Non-sedentary time during awake hours during leisure, hrs/week | 30.5 (25.6–32.8) | 32.5 (27.6–40.2) | 30.6 (26.5–39.2) |
| Leisure time (excluding sleep), hrs/week | 84.9 (77.8–89.6) | 80.2 (74.7–83.8) | 82.7 (76.8–86.2) |
| *Children* | | | |
| N | 7* | 7* | 14* |
| Age, yrs | 9 (6–10) | 9 (7–10) | 9 (7–10) |
| Gender, % female | 28.6 | 28.6 | 28.6 |
| Smartphone and tablet usage during leisure, hrs/week | 12.8 (6.3–31.6) (N = 3) | 2.4 (0.3–6) (N = 5) | 6.1 (1.4–10.9) (N = 8) |
| Smartphone and tablet usage after six pm, hrs/week | 1.4 (1.4–7.5) (N = 3) | 0.2 (0–3.5) (N = 5) | 1.4 (0.1–3.9) (N = 8) |
| Non-sedentary time during awake hours during leisure, hrs/week | 20.4 (15.2–24.5) | 23.2 (19.5–25.9) | 20.6 (19.2–24.5) |
| Leisure time (excluding sleep), hrs/week | 66.9 (55.5–70) | 67.5 (63.9–69.3) | 67.2 (63.9–69.3) |
| | Family/household level | | |
| n | 6 | 6 | 12 |
| Adults per family, n | 2 (1–2) | 1.5 (1–2) | 1 (1–2) |
| Children per family, n | 1 (1–1) | 1 (1–1) | 1 (1–1) |
| Smartphone and tablet usage during leisure, hrs/week | 16.8 (15.1, 49.2) | 15.3 (10.8, 20) | 16.8 (11.4, 31.1) |
| Smartphone and tablet usage after six pm, hrs/week | 6.2 (5.4, 18.4) | 6.7 (6.4, 10) | 6.5 (5.8, 14.2) |
| TVs, n | 1 (1–2) | 1 (1–2) | 1 (1–2) |
| TV output, hrs/week | 39.3 (21.3, 59.6) | 22.3 (0, 27.4) | 26.8 (16, 39.3) |
| TV output after six pm, hrs/week | 21.1 (13.5, 25.1) | 11 (0, 20.3) | 17.1 (6.3, 21.8) |
| Total screen media use output after six pm, hours/week | 26.9 (16.2, 46.8) | 17.6 (10, 28.1) | 23.2 (12.6, 33.6) |
| Total leisure screen media use output, hours/week | 54.6 (31.4, 111.6) | 34.4 (20, 45.9) | 42.4 (29.3, 66.2) |
| Total screen media use output, hours/week | 60.8 (36.2, 114.8) | 37.9 (27.8, 48.9) | 45.8 (31.5, 72.5) |

The table gives an overview of the baseline characteristics of the sample, including baseline screen media use, presented for children and adults separately, within each group, as well as for the whole family or household (aggregation of data). Note that because only three individuals had their own TV, TV-usage at an individual level, was not summarized.

*n is reflective of the whole group for each variable, unless otherwise specified in the table cell.

Medians with 25th and 75th percentiles for all continuous variables, and proportions for all variables that are categorically scaled, are presented. Leisure time was computed based on schedules completed each day during baseline, by subtracting sessions of sleep and work/school, from the total duration of baseline (7 complete days spanning eight days). Note that this table only includes active participants and thus 'household' statistics do not include e.g. younger or older siblings. ISCED: International Classification of Education.

**Table 2. Overview of screen media use and compliance rates during two-week interventions.**

| Individual level (children and adults) | |
|---|---|
| *Evening Restrict* | |
| N | 14** |
| Total screen media use after six pm, hrs/2 weeks | 2.8 (1–4.6) |
| Compliant, % (n) | 0 (n = 0) |
| Residual TV-time, hrs/2 weeks* | 1.2 (0.3, 2.2) |
| *General Restrict* | |
| N | 15** |
| Total leisure entertainment-based screen media use, hrs/2 weeks | 5.1 (3.8, 7.5) |
| Amount of screen media use compared to compliance threshold, %*** | 84.2 (62.5, 125.4) |
| Compliant, % (n) | 66.7 (n = 10) |
| Residual TV-time, hrs/2 weeks* | 1.2 (0, 4) |
| Family/household level | |
| *Evening Restrict* | |
| N | 6 |
| Total screen media use after six pm, hrs/2 weeks | 8.6 (5.7, 11.6) |
| Compliant, % (n) | 0 (n = 0) |
| *General Restrict* | |
| N | 6 |
| Total leisure entertainment-based screen media use, hrs/2 weeks | 18.2 (10.5, 23.2)**** |
| Amount of screen media use compared to compliance threshold, %*** | 97 (72.8, 128.6) |
| Compliant, % (n) | 50 (n = 3) |

This table gives an overview of screen media use totaled during the 2-week intervention as well as corresponding compliance rates. To the extent that it was possible, individual level statistics were given, as well as aggregated family-level statistics. Because the data in the table is described per two weeks, i.e. the duration of the interventions, one should divide by two, to get the data expressed per week. By doing so, comparisons can be made to baseline levels (see Table 1). Summary statistics above are medians with 25th and 75th percentiles.

*Residual TV-time refers to TV-time on shared TVs during the intervention period, which could not by assigned a personal user. It does, therefore, not pertain to any individuals, even though it is summarized under the 'Individual level' headline. Note that residual TV-time in the Evening Restrict group refer only to TV-time after six pm. Residual TV-time is included in the aggregate, family/household level statistics.

**No screen media use data assessment was possible in three individuals from the Evening Restrict group and one individual in the General Restrict group.

***Note that a percentage is calculated for each individual/family and therefore, the percentage is treated as a continuous variable.

****Note that this statistic is included for descriptive purposes and compliance cannot be assessed based on this number alone (due to variations in family size and thus amount of permitted screen media use).

## Potential of interventions to impact children's non-sedentary time

Fig 4 illustrates for the children the individual and group changes in non-sedentary time (min/day).

Among children in the Evening Restrict group, there appeared to be a slight increase in mean non-sedentary time from baseline to follow-up (mean (SE): 15.6 (13.1) min/day, N = 7). In the General Restrict group, one child was excluded from analysis due to invalid accelerometry data at follow-up. Of the six children remaining, all but one child increased their non-sedentary time and there was a mean increase in non-sedentary time (mean (SE): 36.6 (23) min/

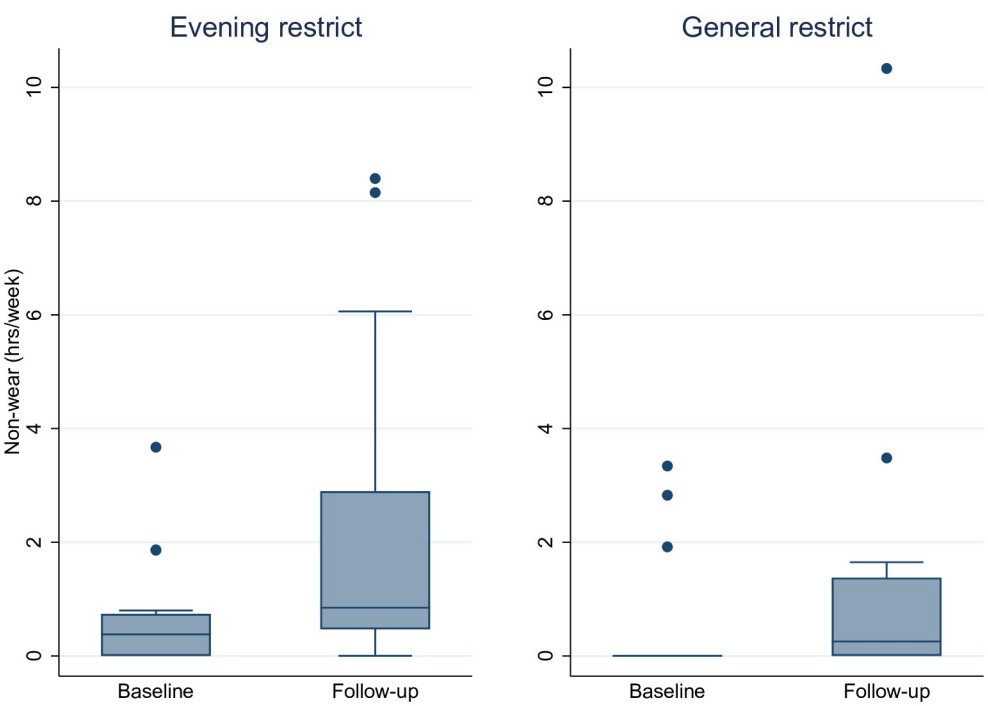

**Fig 3. Graphical overview of compliance to accelerometry during leisure (excluding sleep) at baseline and follow-up.** The figure above illustrates the total number of hours of non-wear at baseline and follow-up, for each intervention group. Baseline and follow-up were each seven days in duration and spanned eight days (the same weekdays twice). Non-wear for adults and children is shown together, above. Hours of non-wear is restricted to only leisure time and during awake hours based on reporting in daily schedules, at both baseline and follow-up.

day). The adults in either group did not appear to change their non-sedentary time from baseline to follow-up (S1 Fig).

## Discussion

The SCREENS pilot trial was carried out with the primary aim to investigate if the approach to intervention was sufficiently successful to reduce children's and adults recreational screen media use. The participants markedly decreased their screen media usage according to their respective interventions. According to the a priori defined strict compliance thresholds of the interventions, no families were compliant in the Evening Restrict group, whereas half were compliant in the General Restrict group. Yet, in the general restrict group the intervention was considered successful considering the large decrease in recreational screen use in all participating families. No participants dropped out of the study after enrollment in either group, which together with the overall successful screen use reduction suggest acceptance of the intervention, in the General Restrict group, particularly. The 12 families were successfully included in the study via the proposed recruitment strategy, which was deemed feasible, and the recruited participants appeared to be comparable to the general pool of eligible families. Compliance to the methodology to assess non-sedentary time based on accelerometry, which is the planned primary outcome in the definitive trial, was high with only one person out of the entire sample, not being sufficiently compliant according to conservative criteria. Lastly, five out of an analytic sample of six children increased their daily non-sedentary time from baseline to follow-up in the General Restrict group, which suggest that this intervention may have the potential

**Table 3. Numerical overview of compliance to accelerometry during leisure (excluding sleep).**

| | Baseline | Follow-up |
|---|---|---|
| **Evening Restrict group** | | |
| N* | 17 | 16 |
| Non-wear, hrs/week | 0.4 (0–0.7) | 0.9 (0.5,2.9) |
| Non-wear, bouts/week | 1 (1, 2) | 2 (2, 3) |
| Valid days, days | 6 (6, 6) | 6 (5,6) |
| Valid weekend days, days | 2 (2, 2) | 2 (2, 2) |
| Compliant**, % | 100 | 100 |
| **General Restrict group** | | |
| N* | 15 | 14 |
| Non-wear, hrs/week | 0 (0, 0) | 0.3 (0, 1.4) |
| Non-wear, bouts/week | 1 (1, 1) | 1 (1, 2) |
| Valid days | 6 (6, 6) | 6 (6, 6) |
| Valid weekend days | 2 (2, 2) | 2 (2, 2) |
| Compliant**, % | 100 | 93.8 |

Summary statistics above are medians with 25th and 75th percentiles.

*This number reflects the number of individuals in each group, at either baseline or follow-up, with recorded non-wear data. If this number is lower than n individuals in each group (N = 17 in Evening Restrict group and N = 16 in General Restrict group) this means that there were some persons with zero non-wear at said time point in said group. Note that because all individuals wore the accelerometers at both points in time, all individuals are included in the statistics regarding valid week- and weekend days, as well as in the proportion who are compliant.

**A valid day of measurement had to include ≤ 10 percent missing data. Furthermore, a complete measurement at baseline and follow-up should include at least four weekdays and at least one weekend day.

to increase non-sedentary in children six-to-ten years of age. The Evening Restrict group appeared to a lesser extent to increase non-sedentary time.

The compliance data should be considered in the context of the demands which were put on the participants. Also, the major decrease in household screen media use, despite non-compliance, should be emphasized. Furthermore, screen media use was registered in more detail during the interventions, compared to during baseline; therefore, the extent of decrease in household screen media use from baseline to intervention most likely is underestimated. The major decrease assures enough contrast in screen use in a future definitive study with a control group. Families in the Evening Restrict group decreased their screen media use after six pm many folds during the intervention. The strict criteria were that absolutely no screen media use after six pm was permitted. The median family screen media usage after six pm during intervention in this group of five families was less than nine hours (or approximately 40 minutes/day). We knew in advance that some families in this intervention arm would be non-compliant to some degree because they had to watch some of the matches of the 2019 World Men's Handball Championship, which Denmark ultimately ended up winning. Also, for this group, some of the non-compliance might simply be explained by e.g. parents intermittently texting or calling their children or family after six pm for reasons such as planning to meet. Although this would be a breach of the rules of the intervention, in retrospect, this should be considered relatively harmless. This does suggest some lack of feasibility in the prescribed intervention in the Evening Restrict group, as currently formulated. In the General Restrict group, compliance rates were moderately high, although screen media use was decreased severalfold. Given the high demands put on the families regarding restrictions of screen media use and given that we can explain some of the instances of non-compliance, the General Restrict arm appears overall

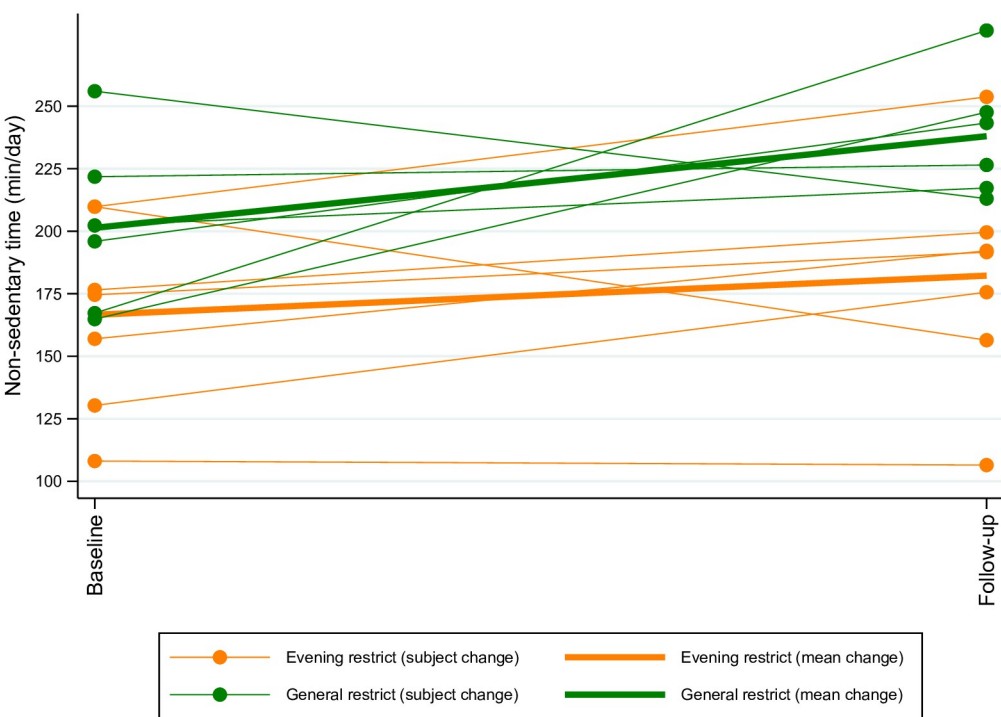

**Fig 4. The potential of two screen media reduction interventions on change in non-sedentary time (min/day) in children six-to-ten years of age.** The figure above illustrates the change in non-sedentary time (min/day) for children six-to-ten years of age. The data is parsed based on group allocation (Evening Restrict group: orange, General Restrict group: green). A thick best fit line for each group has been added to display direction of change in mean non-sedentary time from baseline to follow-up.

to be feasible. Furthermore, some of the issues which could explain non-compliance, could be mitigated simply by timing the interventions better, and being slightly less liberal during the recruitment process. It is the researchers' impression that General Restrict arm was more favorably received compared to the Evening Restrict group.

The results from the assessments of accelerometry showed very high levels of compliance, which suggest very high levels of feasibility, in both children and adults. Notably, the compliance rates should be considered in the context of the families also having to collect additional data using other instruments, e.g. saliva sampling during the morning routine and sleep monitoring. Compared to many other studies, especially large-scale observational studies, where non-compliance to accelerometry is a well-known obstacle, our non-compliance rates were an unprecedented low. One of the main explanations most likely lies in the fact that the 14 days of accelerometry were within a very short time-period, where the study participants met with the researchers three times. Therefore, compared to investigations where participants to a larger extent are left on their own without much researcher contact, here, when followed more closely, participants appear to be encouraged to be compliant. Also, the fact that each family was engaged in the project to a large extent as a family unit, giving and receiving support during the data collection, could also partly explain the high rates of compliance to accelerometry.

There appeared to be a marked and slight increase in mean non-sedentary time, in children in the General Restrict and Evening Restrict group, respectively. One of the main explanations for the group difference is most likely that children in the Evening Restrict group had limited

possibility to replace screen use with activities that were non-sedentary considering the expected early bedtime of children at this age. Therefore, it is expected, given the differences in how the two interventions target children's everyday life that the General Restrict group would potentially induce a larger change in non-sedentary time. Based on data from the randomized controlled trial, which will include a control group and a sample of appropriate size for statistical hypothesis testing, analyses will be conducted to examine the effect size of efficacy.

## Strengths and limitations

The results of the study should be interpreted in the context of its strengths and limitations. A major strength of the study is the use of objective measurements of TV, smartphone, and tablet activity, which overcomes some of the limitations of assessment of screen media use via questionnaires. Moreover, device-based assessment of non-sedentary time with a possibility of objective detailed monitoring of compliance is also a major strength. Furthermore, using a sampling- and recruitment approach based on a well-defined probability-based source population is also a major strength. One of the main limitations is the fact that we did not monitor computer (stationary or laptop) activity—a recognized source of entertainment-based screen media use. Although this in a few instances was reported in the screen media sheets and therefore could be added to total screen media use, proper data on computer use was lacking. Furthermore, we did not include a sheet in which families should systematically report necessary screen media use. This was sometimes reported as an additional note; however, this was to a very limited extent. Because adult participants in the General Restrict group were told that they were allowed up to 30 minutes per day of necessary screen media use, and because this was not noted anywhere, some screen media use recorded during the intervention on smartphones and tablets could be misclassified as entertainment-based (and potentially as non-compliance), when it was in fact necessary screen media use. Also, it is possible that some residual TV-time was usage solely by 'passive participants', which would lead to an overestimation of non-compliance rates. Another issue is that some of our internal work suggest that the SDU DT application may to some extent underreport smartphone and tablet usage. However, by combining objective and subjective data, we hope that we to some extent might overcome this problem. Importantly, only one other published screen-media reduction randomized controlled trial has tried to monitor the exposure objectively [9], and thereby document compliance with greater accuracy and confidence. We can never safeguard against underreporting of screen media use on units that we did not monitor, e.g. screen media use in other households. However, it is the researchers' impression that the participants were highly engaged in the project, which there is no substitute for. Another limitation is that because we made several changes to the families' everyday life, including asking them to systematically write down screen media use and to generally reduce screen media consumption, we may not be able to pinpoint which initiative was more effective (if any).

## Future perspectives

The data collected and the experiences acquired from the pilot study has led to several additions or changes to the definitive trial protocol. Firstly, due to the limitations of the Evening Restrict group protocol and the strengths of the General Restrict intervention (including its apparent feasibility), the full-scale randomized controlled trial only includes the latter arm, and a control group. Secondly, to improve the quality of the data and subsequent analyses, we have developed software to monitor computer usage, as well as created a sheet in which necessary screen media use can be reported. Also, a checklist has been created, which must be placed next to all common TV's during baseline. On this checklist, each family member must mark

each time they use the TV, such that a baseline profile of users of the TV can be defined. Lastly, to quality check the SDU DT data we will cross-reference our IOS smartphone and tablet data with iOS's internal screen media use summarizer (Screen Time function integrated in iOS 12 and more recent versions). Another future perspective pertains to scheduling of the trial. If there are upcoming TV-events or other scenarios, where the families cannot restrict screen media use, we will postpone the whole trial, to a time that is more suitable. Lastly, a major change is that in the future we will include siblings four and five years, as well as siblings between 11–14 years as participants. The reason for this change is twofold. First, it was our experience that because older and younger siblings were excluded, this might have affected the feeling of being engaged as a whole family. In fact, we noted that two families chose not to participate because engagement in the project would only be for some members of the family. Secondly, because it was our experience that the participants were highly engaged, we would expect this to also be the case for the additional siblings, if included. Even though our reservations regarding the children 11–14 years of age still to some extent hold true, we believe that the benefits of including them by far outweigh the limitations of excluding them.

We believe by incorporating field-experience with implementation of the trial, with state-of-the-art measurements of intervention compliance and outcomes, we will be able to implement a successful full-scale SCREENS randomized controlled efficacy trial. We expect that our future findings will break new ground in our understanding of the effect of restricting screen media use on physical activity, sleep, and physiological stress in families with children.

## Conclusions

Our findings showed that degree of compliance was sufficiently high for the General Restrict group but was unsatisfactory for the Evening Restrict group. However, some issues relating to non-compliance were expected, and several steps can be taken to mitigate these in a definitive full-scale randomized controlled trial. The suggested large within-group decrease in recreational screen use in the General restrict group and the zero percent drop-out deem it acceptable to proceed with in a full definitive trial with a control group. Compliance to the accelerometry assessment, which is the basis for the planned primary outcome in the definitive trial, was high with almost 100% compliance. The survey-based recruitment strategy was feasible and secured enough study participants. Also, the strategy allowed for comparisons between study participants and eligible and ineligible individuals from a well-defined source population.

## Supporting information

**S1 Checklist. The CONSORT extension.**
(DOC)

**S1 File.**
(PDF)

**S1 Fig. The potential of two screen media reduction interventions on change in non-sedentary time (min/day) in adults.** The figure above illustrates the change in non-sedentary time (min/day) for adults participating in the SCREENS pilot trial. The data is parsed based on group allocation (Evening Restrict group: orange, General Restrict group: green). A thick best fit line for each group has been added to display direction of change in mean non-sedentary time from baseline to follow-up, in each group.
(TIF)

**S1 Table. Characteristics of participants from the source population, survey non-respondents, survey respondents who were ineligible or eligible to participate in the feasibility trial based on initial criteria, and feasibility trial participants.**
(DOCX)

## Acknowledgments

We would like to greatly thank the twelve families who participated in the pilot study, whose dedication and time was essential to developing this paper.

We would also like to thank the researcher service organization Open Patient data Explorative Network for their work, which included sending an electronic survey and cover letter, as well as managing the storage of data under conditions which comply with the General Data Protection Regulation.

We would also like to thank Henrik Olsen and Kristian Jacobsen of the engineer staff at the Department of Sports Science and Clinical Biomechanics, for the many hours spent creating and testing the TV monitoring devices. We would furthermore like to thank the student helper for typing in reporting from checklists and screen media diaries.

Fig 2 is a re-use of Fig 3 from Rasmussen, M.G.B., Pedersen, J., Olesen, L.G. *et al*. Short-term efficacy of reducing screen media use on physical activity, sleep, and physiological stress in families with children aged 4–14: study protocol for the SCREENS randomized controlled trial. *BMC Public Health* **20**, 380 (2020). https://doi.org/10.1186/s12889-020-8458-6. The re-use is permitted under the Creative Commons By Attribution license. A small pink arrow was added to the original figure, which illustrates that a questionnaire was administered following baseline assessment.

## Author Contributions

**Conceptualization:** Martin G. B. Rasmussen, Jesper Pedersen, Line Grønholt Olesen, Peter Lund Kristensen, Jan Christian Brønd, Anders Grøntved.

**Data curation:** Martin G. B. Rasmussen, Jesper Pedersen, Line Grønholt Olesen, Peter Lund Kristensen, Jan Christian Brønd, Anders Grøntved.

**Formal analysis:** Martin G. B. Rasmussen.

**Funding acquisition:** Anders Grøntved.

**Methodology:** Martin G. B. Rasmussen, Jesper Pedersen, Line Grønholt Olesen, Peter Lund Kristensen, Jan Christian Brønd, Anders Grøntved.

**Supervision:** Anders Grøntved.

**Writing – original draft:** Martin G. B. Rasmussen.

**Writing – review & editing:** Martin G. B. Rasmussen, Jesper Pedersen, Line Grønholt Olesen, Peter Lund Kristensen, Jan Christian Brønd, Anders Grøntved.

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
