## [Decision Letter · Decision Letter 0]

14 Jul 2021

PONE-D-21-11116

Feasibility of two screen media reduction interventions: results from the SCREENS pilot trial

PLOS ONE

Dear Dr. Rasmussen,

Thank you for submitting your manuscript to PLOS ONE. After careful consideration, we feel that it has merit but does not fully meet PLOS ONE’s publication criteria as it currently stands. Therefore, we invite you to submit a revised version of the manuscript that addresses the points raised during the review process.

We look forward to receiving your revised manuscript.

Kind regards,

Walid Kamal Abdelbasset, Ph.D.

Academic Editor

PLOS ONE

Journal Requirements:

2. Thank you for submitting your clinical trial to PLOS ONE and for providing the name of the registry and the registration number. The information in the registry entry suggests that your trial was registered after patient recruitment began. PLOS ONE strongly encourages authors to register all trials before recruiting the first participant in a study.

1) your reasons for your delay in registering this study (after enrolment of participants started);

2) confirmation that all related trials are registered by stating: “The authors confirm that all ongoing and related trials for this drug/intervention are registered”.

Reviewers' comments:

Reviewer's Responses to Questions

**Comments to the Author**

1. Is the manuscript technically sound, and do the data support the conclusions?

Reviewer #1: No

Reviewer #2: Yes

2. Has the statistical analysis been performed appropriately and rigorously? 

Reviewer #1: No

Reviewer #2: Yes

3. Have the authors made all data underlying the findings in their manuscript fully available?

Reviewer #1: No

Reviewer #2: Yes

4. Is the manuscript presented in an intelligible fashion and written in standard English?

Reviewer #1: No

Reviewer #2: Yes

5. Review Comments to the Author

Reviewer #1: The objective of this 2-arm, parallel group, pilot, randomized trial (SCREENS) is to assess compliance and feasibility of the interventions, (i) evening restriction, and (ii) General restriction, in a Danish sample. The study was registered as a RCT within the clinicaltrials.gov (with a valid NCT #), and was approved by the respective IRB/Ethics Committee. While the study objectives sound interesting, is important, and on target, some shortcomings were observed, in regards to abiding by the CONSORT guidelines for conducting and reporting results of high-quality RCTs. Some other (statistical) comments were also provided.

1. Methods:

Methods reporting require an orderly manner following CONSORT guidelines, without repeating information, such as Trial Design, Participant Eligibility criteria and settings, Interventions, Outcomes, sample size/power considerations, Interim analysis and stopping rules, Randomization (details on random number generation, allocation concealment, implementation), Blinding considerations, should be mentioned explicitly. The authors are advised to create separate subsections for each of the possible topics (whichever necessary), and that way produce a very clear writeup. I see the Authors already made a sincere attempt; however, they are advised to write it carefully, following nice examples in the manuscript below:

https://www.sciencedirect.com/science/article/pii/S0889540619300010

Specific comments:

(a) For instance, the randomization and allocation concealment should be made very clear (they are NOT the same thing); the trial staff recruiting patients should NOT have the randomization list. Randomization should be prepared by the trial statistician, and he/she would not participate in the recruiting.

(b) Block randomization is often recommended in (pilot) trials, to ensure a balance in sample size across groups over time. The writeup suggests "alternating blocks of 2 or 4 families" were used; do they mean blocl randomization here? Make it clear.

https://www.ncbi.nlm.nih.gov/pmc/articles/PMC2267325/

(c) Sample size/power: I do not necessarily agree that a sample size justification is not needed for a planning/pilot trial. See paper linked below. Can a post-hoc sample size computations be provided, so that readers have a proper idea on what they can expect, and/or what effect size the investigators wanted to base their study upon? Just choosing a number (like ~ 12 here) is not a well-thought-out science.

https://www.ncbi.nlm.nih.gov/pmc/articles/PMC4876429/

(d) Statistical Analysis: This is a major limitation. The statistical methods section do not state clearly what methods will be employed to assess differences, and testing thus conducted at what level of significance.

2. Results & Conclusions:

(a) Analysis reported appeared half-baked. Only larger/smaller increase (in mean non-sedentary time and standard errors) were reported, without considering any formal "statistical testing", via various available tests. I understand this is a feasibility study reporting; still, statistical tests are required to present inference on comparing groups.

(b) The Conclusions section should clearly state the very pilot nature of the study, using N = 12. Again, this is a pilot study, but when the researchers have access to N = 1675, I do not see much justification is selecting N = 12!

Reviewer #2: Introduction

need editing and rewrite

Methods:

Kindly focus on three basic elements of methods section.

How the study was designed?

How the study was carried out?

How the data were analyzed?

Eligibility criteria for participants not clear and Provide sufficient details of interventions of each group to allow replication

Statistics

please rewrite again and discuss which test you used and why also all tables need rearrange

Outcomes and estimation need to be explained well

Discussion:

needs to be as per well defined objectives

Describe sources of potential bias and imprecision

It has to be framed in such a way that readers are able to have good understanding of the current evidences and rationale of the paper

6. PLOS authors have the option to publish the peer review history of their article (what does this mean?). If published, this will include your full peer review and any attached files.

Reviewer #1: No

Reviewer #2: No

---

## [Author Response · Author response to Decision Letter 0]

30 Sep 2021

Review Comments to the Author

Reviewer #1: The objective of this 2-arm, parallel group, pilot, randomized trial (SCREENS) is to assess compliance and feasibility of the interventions, (i) evening restriction, and (ii) General restriction, in a Danish sample. The study was registered as a RCT within the clinicaltrials.gov (with a valid NCT #), and was approved by the respective IRB/Ethics Committee. While the study objectives sound interesting, is important, and on target, some shortcomings were observed, in regards to abiding by the CONSORT guidelines for conducting and reporting results of high-quality RCTs. Some other (statistical) comments were also provided.

Response:

Thank you for the comments, we will address them point by point below.

1. Methods:

Methods reporting require an orderly manner following CONSORT guidelines, without repeating information, such as Trial Design, Participant Eligibility criteria and settings, Interventions, Outcomes, sample size/power considerations, Interim analysis and stopping rules, Randomization (details on random number generation, allocation concealment, implementation), Blinding considerations, should be mentioned explicitly. The authors are advised to create separate subsections for each of the possible topics (whichever necessary), and that way produce a very clear writeup. I see the Authors already made a sincere attempt; however, they are advised to write it carefully, following nice examples in the manuscript below:

https://www.sciencedirect.com/science/article/pii/S0889540619300010

Response:

We agree that this is very helpful for a reader of the paper. We have revised the structure of the methods section to better follow the suggested CONSORT structure. Thank you for the nice example. We have also removed unnecessary or repeated information. 

Specific comments:

(a) For instance, the randomization and allocation concealment should be made very clear (they are NOT the same thing); the trial staff recruiting patients should NOT have the randomization list. Randomization should be prepared by the trial statistician, and he/she would not participate in the recruiting.

Response:

We have updated the Randomization (random number generation, allocation concealment, implementation) subsection. It now reads:

“Families were randomized in 1:1 allocation ratio to one of two interventions using random permuted blocks of two or four families, stratifying according whether the child was an only child. The randomization was performed using a researcher service available in the Region of Southern Denmark; Odense Patient Network Randomize. The random allocation sequence was created by staff at Odense Patient Network Randomize and they were not involved with recruitment of participants. The sequence was concealed to the researchers, and when the individual family was randomized, it was done by a researcher in the home of the participants by logging on to the Odense Patient Network Randomize web application.”

(b) Block randomization is often recommended in (pilot) trials, to ensure a balance in sample size across groups over time. The writeup suggests "alternating blocks of 2 or 4 families" were used; do they mean blocl randomization here? Make it clear.

https://www.ncbi.nlm.nih.gov/pmc/articles/PMC2267325/

Response:

We agree that this was unclear. As you can read above, we have updated the section. 

(c) Sample size/power: I do not necessarily agree that a sample size justification is not needed for a planning/pilot trial. See paper linked below. Can a post-hoc sample size computations be provided, so that readers have a proper idea on what they can expect, and/or what effect size the investigators wanted to base their study upon? Just choosing a number (like ~ 12 here) is not a well-thought-out science.

https://www.ncbi.nlm.nih.gov/pmc/articles/PMC4876429/

Response:

We did not carry out a sample size calculation as we had no aim to incorporate hypothesis testing in the feasibility trial. The sample size would be expected to be of insufficient size for this purpose. The main aims of conducting the feasibility trial were to examine compliance to the trial interventions, participant compliance to the planned definitive trial outcome assessment protocol, and the feasibility of a strategy to recruit participants into the study. These aims of the pilot trial were decided to safeguard internal validity for making inferences about efficacy in future definitive trial and to field-test logistical aspects of the future definitive trial. According to CONSORTs own guidelines for feasibility studies, inclusion of an objective to test a hypothesis of effectiveness (or efficacy) is not recommended and any estimates of effect using participant outcomes, as they are likely to be measured in the future definitive RCT, should be reported as estimates with 95% confidence intervals without P values—because pilot trials are not powered for testing hypotheses about effectiveness (Eldridge SM, Chan CL, Campbell MJ, Bond CM, Hopewell S, Thabane L, et al. CONSORT 2010 statement: extension to randomised pilot and feasibility trials. BMJ. 2016;355:i5239).

The paper by Whitehead et al. (https://www.ncbi.nlm.nih.gov/pmc/articles/PMC4876429/) provides a suggestion to set the sample size of a pilot trial when the primary aim of the pilot being to estimate the standard deviation of the definitive trial outcome sample size calculation. Because we had no specific aim to use the pilot trial to obtain reliable estimates of standard deviation of the planned definitive trial primary outcome (we obtained this information elsewhere), we do not justify our sample size based on calculation. In line with CONSORT we provide our rationale for inclusion of 12 families with at least one child and one adult, with an expected total sample size of more than 30 participants. We deemed that this was sufficient to investigate the primary aims of the feasibility trial.

(d) Statistical Analysis: This is a major limitation. The statistical methods section do not state clearly what methods will be employed to assess differences, and testing thus conducted at what level of significance.

Response:

We refer to our response above. We report within-group means and standard errors or other descriptive statistics for data that were not normally distributed, which is in line with CONSORTs own guidelines for feasibility trials. 

2. Results & Conclusions:

(a) Analysis reported appeared half-baked. Only larger/smaller increase (in mean non-sedentary time and standard errors) were reported, without considering any formal "statistical testing", via various available tests. I understand this is a feasibility study reporting; still, statistical tests are required to present inference on comparing groups.

Response:

We refer to our response above.

(b) The Conclusions section should clearly state the very pilot nature of the study, using N = 12. Again, this is a pilot study, but when the researchers have access to N = 1675, I do not see much justification is selecting N = 12!

Response:

We can see that our rationale for this approach needs some more explanation. A challenge for most randomized trials is to generalize results beyond the study population. An important aspect of our strategy of recruitment was that we wanted to be able to better judge the extent that study findings could be applied to a well-defined source population (here Danish families with children aged 6-10-years of age from the general population). Furthermore, another aim of our population-based survey that we used for the recruitment, was to provide new descriptive data on screen media habits among families with children and investigate whether several putative proximal correlates are associated with children’s screen use. This latter aim is not part of the current paper. As we describe in detail in the flow chart (Figure 1), we distributed our survey to 1,675 randomly selected parents sharing address with a child 6-10-ears of age. Sampling of these participants were based on the Central Person Register in Denmark, in which each person has their own social security number. We obtained information on date of birth and sex on the selected child and adult (all N=1,675 dyads) from the Danish Health Data Authority. Of these 394 (23.4%) responded to the survey, which was an expected response rate considering the age of the parents. In the definitive trial we also negotiated with the Danish Health Data Authority to also receive information on ethnicity of child and parent, however, this was unfortunately not part of the background information on the source population based on the random sample in the pilot trial. By having available background information from all individuals from the source population and detailed information from the survey on a sub-sample of the source population (the respondents), our goal in the definitive trial is to be able to much better judge whether the results of the future definitive trial could be applied to 1) the eligible population of families with children and 2) the source population. Thus, evaluation of the extent that we succeeded with this was also an aim in the feasibility trial. The alternative sampling methodology is a nonprobability sampling by convenience, e.g. via social media platforms, and using this approach we have very limited possibility to examine the possibility to generalize (or judge generalizability) the results beyond the study population itself. Hence, we decided to recruit based on a population-based probability sampling, which in our view could be a major strength of the future definitive trial. 

Going from the N=1,675 to N=12 participating families, it is important to recognize the details of the flow. Firstly, it is important to discriminate between the N=394 respondents and the respondents being eligible to the feasibility trial based on the initial screening of eligibility. Here, N=90 of the N=394 families were deemed initially eligible for participation in the feasibility trial (see detailed criteria for inclusion and exclusion in the methods section of the manuscripts). Of these, N=21 was initially interested and also deemed eligible according to the secondary criteria that was screened during a telephone interview, however, N=9 of these ended up not participating due to several reasons (see Figure 1 flow chart). Again, this strategy of sampling should be viewed in light of a plan to use a similar approach in the definitive trial with an expected 10-fold more families participating. To further explain the rationale for the approach to sampling, we have added this to the ‘Participants, eligibility criteria, and settings’ subsection:

“The strategy of sampling of participants into the study was designed to optimize the possibility to generalize (or better judge generalizability of) the study findings in a future definitive trial to a well-defined source population (Danish families with children aged 6-10-years of age from the general population)”

Furthermore, we have added an additional table (Supplementary Table 1) comparing important characteristics between of participants from the source population, non-respondents, respondents being ineligible to participate, respondents being eligible to participate in the feasibility trial based initial criteria, and participants in feasibility trial. We have also commented on this in the results section and in the discussion section of the revised paper in the context of the aim of the paper to examine the feasibility of a strategy to recruit participants into the study. 

Reviewer #2: Introduction

need editing and rewrite

Methods:

Kindly focus on three basic elements of methods section.

How the study was designed?

How the study was carried out?

How the data were analyzed?

Response:

This was also a comment from reviewer 1. We have revised the structure of the methods section to better follow the suggested CONSORT structure. We have also removed unnecessary or repeated information. 

Eligibility criteria for participants not clear and Provide sufficient details of interventions of each group to allow replication

Response:

We have provided these eligibility criteria in the paper:

Families were initially eligible to participate in the trial, if they met the following inclusion criteria, assessed via questions included in the survey:

- The adult respondent had to be above the 50th percentile for weekly screen media use amount. In absolute terms, this amounted to consuming above 2 hours and 43 minutes of leisure screen media use on a typical day of the week (based on all survey respondents). For practical reasons pertaining to recruitment, we chose to limit this criterion only to adults, whose screen media use might be a decent proxy for the household screen media use consumption. 

- Households should not include children less than four years of age. This was to avoid disturbance of data collection on sleep quality and duration (an included outcome measurement) because of infants and toddlers with irregular sleep habits.

Furthermore, families had to meet the following secondary inclusion criteria, which was assessed during telephone screening:

- At least one adult and one child between six and ten years of age had to participate in the study

- All actively participating family members had to be able to remove both leisure- and work-based screen media use during evenings and during weekends

- Families had to consider the extent of their screen media usage an issue and report to be motivated to decrease leisure screen media use for a short time-period

- Lastly, the families had to declare that passive participants would respect the intervention conditions that those family members, who were active participants, had to follow

We excluded participants during telephone screening based on the following criteria:

- If children only resided part-time in the household

- If any participants had been diagnosed with stress or a sleep disorder by their general practitioner within the last 12 months

- If adults regularly worked night shifts

- If family members were unable to do basic physical activity during everyday life

- If family members were diagnosed with or in the process of getting cleared from developmental disorders, such as autism spectrum disorders, or neuropsychiatric disorders, such as attention deficit hyperactivity disorder 

- Lastly, if family members were already actively participating in a research study, e.g. the Odense Child Cohort study (an ongoing population-based birth study in Odense, Denmark)

We have tried to include some more information about the intervention in the revised paper.

Statistics

please rewrite again and discuss which test you used and why also all tables need rearrange

Outcomes and estimation need to be explained well

Response:

We have added some more details to the statistics section. Please note that we did not carry out any hypothesis testing in the feasibility trial. The sample size would be expected to be of insufficient size for this purpose. The main aims of conducting the feasibility trial were to examine compliance to the trial interventions, participant compliance to the planned definitive trial outcome assessment protocol, and the feasibility of a strategy to recruit participants into the study. These aims of the pilot trial were decided to safeguard internal validity for making inferences about efficacy in future definitive trial and to field-test logistical aspects of the future definitive trial. According to CONSORTs own guidelines for feasibility studies, inclusion of an objective to test a hypothesis of effectiveness (or efficacy) is not recommended (Eldridge SM, Chan CL, Campbell MJ, Bond CM, Hopewell S, Thabane L, et al. CONSORT 2010 statement: extension to randomised pilot and feasibility trials. BMJ. 2016;355:i5239). Thus, the statistics we report in the paper is descriptive. We have also added this to the statistics section in the revised paper. 

Discussion:

needs to be as per well defined objectives

Describe sources of potential bias and imprecision

It has to be framed in such a way that readers are able to have good understanding of the current evidences and rationale of the paper

Response:

Some more explaining details have been added to the discussion section. In particular, the first section of the discussion and the conclusion have been framed in a way to help the reader understand the results in the context of the aims of the feasibility trial. Also, sources of strengths and limitations section has been updated in the revised manuscript

---

## [Decision Letter · Decision Letter 1]

25 Oct 2021

Feasibility of two screen media reduction interventions: results from the SCREENS pilot trial

PONE-D-21-11116R1

Dear Dr. Rasmussen,

We’re pleased to inform you that your manuscript has been judged scientifically suitable for publication and will be formally accepted for publication once it meets all outstanding technical requirements.

Kind regards,

Walid Kamal Abdelbasset, Ph.D.

Academic Editor

PLOS ONE

Additional Editor Comments (optional):

Reviewers' comments:

Reviewer's Responses to Questions

**Comments to the Author**

1. If the authors have adequately addressed your comments raised in a previous round of review and you feel that this manuscript is now acceptable for publication, you may indicate that here to bypass the “Comments to the Author” section, enter your conflict of interest statement in the “Confidential to Editor” section, and submit your "Accept" recommendation.

Reviewer #1: All comments have been addressed

Reviewer #2: All comments have been addressed

2. Is the manuscript technically sound, and do the data support the conclusions?

Reviewer #1: (No Response)

Reviewer #2: Yes

3. Has the statistical analysis been performed appropriately and rigorously? 

Reviewer #1: (No Response)

Reviewer #2: Yes

4. Have the authors made all data underlying the findings in their manuscript fully available?

Reviewer #1: (No Response)

Reviewer #2: Yes

5. Is the manuscript presented in an intelligible fashion and written in standard English?

Reviewer #1: (No Response)

Reviewer #2: Yes

6. Review Comments to the Author

Reviewer #1: (No Response)

Reviewer #2: (No Response)

7. PLOS authors have the option to publish the peer review history of their article (what does this mean?). If published, this will include your full peer review and any attached files.

Reviewer #1: No

Reviewer #2: No

---

## [Editor Report · Acceptance letter]

3 Nov 2021

PONE-D-21-11116R1 

Feasibility of two screen media reduction interventions: results from the SCREENS pilot trial 

Dear Dr. Rasmussen:

I'm pleased to inform you that your manuscript has been deemed suitable for publication in PLOS ONE. Congratulations! Your manuscript is now with our production department. 

Kind regards, 

on behalf of

Dr. Walid Kamal Abdelbasset 

Academic Editor

PLOS ONE